# Multiple Chemical Sensitivity in Patients Exposed to Moisture Damage at Work and in General Working-Age Population—The SAMDAW Study

**DOI:** 10.3390/ijerph182312296

**Published:** 2021-11-23

**Authors:** Pia Nynäs, Sarkku Vilpas, Elina Kankare, Jussi Karjalainen, Lauri Lehtimäki, Jura Numminen, Antti Tikkakoski, Leenamaija Kleemola, Heini Huhtala, Jukka Uitti

**Affiliations:** 1Faculty of Medicine and Health Technology, Tampere University, 33520 Tampere, Finland; jussi.karjalainen@pshp.fi (J.K.); lauri.lehtimaki@tuni.fi (L.L.); leenamaija.kleemola@gmail.com (L.K.); jukka.uitti@tuni.fi (J.U.); 2Department of Phoniatrics, Tampere University Hospital, 33520 Tampere, Finland; sarkku.vilpas@pshp.fi (S.V.); eliina.kankare@pshp.fi (E.K.); 3Allergy Centre, Tampere University Hospital, 33520 Tampere, Finland; jura.numminen@pshp.fi; 4Department of Clinical Physiology and Nuclear Medicine, Tampere University Hospital, 33520 Tampere, Finland; antti.tikkakoski@pshp.fi; 5Faculty of Social Sciences, Tampere University, 33520 Tampere, Finland; heini.huhtala@tuni.fi

**Keywords:** multiple chemical sensitivity (MCS), chemical intolerance, moisture damage, mold, dampness

## Abstract

A considerable proportion of patients having respiratory tract or voice symptoms associated with workplace moisture damage (MD) could have multiple chemical sensitivity (MCS). MCS is characterized by symptoms of different organ systems in association with low-level chemical exposure. The objective of this study was to assess the prevalence of MCS among patients referred to secondary health care because of respiratory or voice symptoms associated with workplace MD compared to the general working-age population. Using three subscales of the QEESI© questionnaire, we assessed MCS in the study patients and 1500 controls in the same district randomly selected from the Finnish Population Information System. Study patients had significantly more often high scores in chemical intolerance (39% vs. 23%, *p* = 0.001), symptom severity (60% vs. 27%, *p* < 0.001), and life impact subscales (53% vs. 20%, *p* < 0.001). Asthma, chronic rhinosinusitis, laryngeal problems, and atopy were not associated with the presence of MCS. MCS is common among patients referred to secondary health care with respiratory tract and/or voice symptoms associated with workplace MD, and it considerably affects their everyday life. MCS should be considered as a possible explanatory factor for MD-associated symptoms.

## 1. Introduction

Multiple chemical sensitivity (MCS) (or chemical/odor intolerance) is a condition characterized by symptoms of different organ systems in association with low-level chemical exposure that is below known harm-causing levels and does not cause symptoms in most people [1]. MCS is a subtype of idiopathic environmental intolerance (IEI) [2], which includes reacting to different environmental factors such as chemicals or odors, electromagnetic fields [3], noise [4], or buildings the person considers “sick” [5]. IEI symptoms cannot be explained by any known toxicological [6], physical [7], or immunological [8,9] mechanisms, but recent studies suggest that central sensitization and change in neurological processing of sensory stimuli could be the key mechanisms causing IEI [10,11,12,13].

A consensus in 1999 set six different criteria for a diagnosis of MCS: the condition is chronic, and symptoms are reproducible, appear in multiple organ systems, occur in response to low-level exposure to different chemicals, and resolve after exposure is ceased [14]. Later, Lacour et al. emphasized the presence of central nervous system symptoms such as headache, fatigue, and cognitive deficits [15]. As there is no recognized biological mechanism explaining MCS, there are no clinical tests for the diagnosis. To screen the presence of MCS, different questionnaires have been developed [16,17,18,19,20] of which the Quick Environmental Exposure and Sensitivity Inventory (QEESI©) [21] seems to be the most widely used [22,23,24,25,26,27,28]. However, there are still no commonly accepted definition and diagnostic criteria for MCS [29].

Epidemiological studies on self-reported MCS during the last decade have presented prevalence between 3% and 26%, being often higher in women than in men [22,28,30,31,32,33,34]. Recent research suggests that MCS perhaps is not as permanent a condition as previously thought [35,36]; Palmquist reported 44% of subjects with specific environmental intolerance (EI) recovering during a six-year follow-up. On the other hand, there was a 13% probability that a certain EI would spread to another type of EI [37]. Regardless, MCS may significantly affect the quality of some subjects’ social and occupational life [23,38].

Previous epidemiological research has concluded that workplace moisture damage (MD) exposure increases the risk of new-onset asthma and respiratory tract symptoms [39,40]. In a clinical setting, only a part of the patients examined in secondary health care for MD-associated symptoms are diagnosed with an organic disease such as asthma [41,42], and a considerable proportion of them seem to have symptoms of different organ systems referring to possible MCS [30,43]. It has also been suggested that non-specific building-related symptoms that cannot be explained by actual indoor air conditions and MCS share partly common symptoms and could be explained with similar mechanisms [13]. However, the possibility of MCS is not routinely examined in patients presenting with symptoms associated with MD exposure. To improve the management of these patients and to evaluate if routine assessment of possible MCS should be part of their diagnostic workup, we need to know the prevalence of MCS in these patients.

The specific objective of this study was to assess how common MCS is among patients referred to secondary health care because of respiratory tract or voice symptoms associated with MD at the workplace compared to the general working-age population.

## 2. Materials and Methods

Patients referred to Tampere University Hospital departments of Occupational Medicine or Phoniatrics or Allergy Centre due to respiratory or voice symptoms associated with MD exposure at workplace were recruited to our study. The study protocol has previously been published in detail [44]. Comprehensive clinical tests were conducted to diagnose possible asthma, chronic rhinosinusitis (CRS), laryngeal problem (dysfunction such as muscle tension in phonation or organic problem such as laryngitis or vocal fold polyp), or atopy (defined by at least one skin prick test positive (≥3 mm) result in standard panel including birch, timothy, mugwort, horse, dog, cat, Dermatophagoides Pteronyssinus house dust mite, and latex) in the patients. The clinical findings of the patients have been presented in a previous article [42]. In addition, patients fulfilled a questionnaire including QEESI© which has been developed for use in research as well as clinical evaluation of patients reporting intolerances [21]. Three QEESI© subscales were used to assess possible MCS: the chemical intolerance subscale to find out which chemicals or odors possibly cause symptoms, symptom severity subscale to examine what kind of and how severe symptoms a person commonly experiences, and life impact subscale to assess how the sensitivities affect different aspects of everyday life (Table 1).

The respondents rated each item in different subscales between 0 and 10 points, 0 meaning not at all a problem and 10 severe or disabling problem. The points of each subscale were tallied to obtain a total score from 0 to 100. In the chemical intolerance and symptom severity subscales, the scores 0–19 were classified as low, 20–39 as medium, and 40–100 as high. In the life impact subscale, the respective scores were 0–11, 12–23, and 24–100. A high score class in the chemical intolerance subscale was used as a criterion for MCS. Based on previous research, this threshold has sensitivity of 83% and specificity of 84% for MCS [21].

To find out if MCS would be more common among the study patients with respiratory tract symptoms associated with MD at workplace than among general working-age population, the same questionnaire was sent to Finnish-speaking controls of the same province with a population of 510,000. Considering the low response rates in surveys nowadays, to obtain a control group of 400 subjects (ratio 4:1), 1500 20–63-year-old persons with proportions of women and men in different age groups equivalent to the study patient population were randomly selected from the Finnish Population Information System. The questionnaire was sent by mail, and a possibility to answer the questionnaire alternatively online was provided.

Independent-samples T-test and chi-squared tests were used to compare categorical and continuous variables between different groups. Data management and analysis were performed using IBM^®^ SPSS^®^ Statistics Version 25 (2017).

The Ethics Committee of the Pirkanmaa Hospital District approved the study (R14095). All the study subjects provided their written informed consent.

## 3. Results

The study patient population recruited between October 2015 and June 2017 consisted of 99 patients, 82 of whom were women and 17 men. Their mean age was 44 years (range 20–63).

The questionnaire was sent to the controls in autumn 2017, and 568 (38%) of them responded, six of them on the internet. The mean age of the controls was 46 years (range 21–63), and 87% of them were women and 13% men. Age, sex, and the proportions of women and men in different age groups did not statistically differ between study patients and controls (data not shown).

### 3.1. Study Patients’ QEESI© Results

Among the study patients, 39% had high scores in chemical intolerance, 60% in symptom severity, and 53% in life impact subscales. The gender difference did not reach statistical significance among the study patients in chemical intolerance (43% and 24%, respectively, *p* = 0.114) or symptom severity subscales (60% and 59%, respectively, *p* = 0.575), but women had high scores more often in the life impact subscale (57% and 29%, respectively, *p* = 0.033).

Among the study patients, 32% had asthma, 39% asthma and/or CRS, 42% laryngeal dysfunction or organic change, and 37% atopy. No statistically significant differences were found in the comparisons of subscale results between patients with and without these conditions (Table 2).

### 3.2. Comparison of QEESI© Results between Study Patients and Controls

The study patients had significantly more often high scores in chemical intolerance (39% vs. 23%, *p* = 0.001), symptom severity (60% vs. 27%, *p* < 0.001), and life impact (53% vs. 20%, *p* < 0.001) subscales than controls (Figure 1). The proportion of subjects scoring high in all the three scales was 26% among the patients and 9% among the controls (*p* < 0.001).

### 3.3. Comparison of QEESI© Results between Women and Men among Population Controls

Among the population controls, women had more often high scores in each of the three subscales compared to men: 25% vs. 10% (*p* = 0.001) in chemical intolerance, 29% vs. 10% (*p* < 0.001) in symptom severity, and 22% vs. 5% (*p* < 0.001) in life impact (Figure 2).

### 3.4. Comparison of QEESI© Results between Population Controls Working and off Work and between Study Patients and Working Controls

Of the population controls, 558 subjects (98%) expressed their employment status: 451 (81%) were currently working and 107 (19%) temporarily (unemployed, students, etc.) or permanently out of work. There were no statistical differences in QEESI© results between those working and off work: they had high scores in the chemical intolerance scale 22% vs. 24% (*p* = 0.268), symptom severity scale 25% vs. 34% (*p* = 0.112), and life impact scale 19% vs. 24% (*p* = 0.349), respectively.

The difference in QEESI© results between working controls and patients (data not shown) was similar to the difference between all the controls and patients presented above. 

## 4. Discussion

This article presents the first study on workplace-MD-exposed patients’ MCS findings compared to the general working-age population. We found that MCS is significantly more prevalent among patients with workplace-MD-associated respiratory tract and/or voice symptoms than among the general population. The most prominent differences between study patients and the general population were in experiencing symptoms and in the effect of sensitivities on different aspects of everyday life.

The prevalence of MCS in the general population was higher in this study (23%) than in the questionnaire study by Karvala et al. (15%) [45]. However, that study was conducted in a certain geographical area in Finland, Ostrobothnia in Western Finland, and the prevalence of self-reported chemical intolerance was assessed with one question. More in line with our study is the study of Vuokko et al. on fertile-age women in Eastern Finland, in which chemical intolerance was determined if the respondent reported intolerance to at least two of the six chemical items asked. Of the respondents, 29% reported annoyance from chemicals without any symptoms and 23% annoyance with one or more symptoms [31]. The prevalence of MCS also varies depending on the target population and on the method and criteria used. Studies with QEESI© on the general population in other countries have resulted in the prevalence of 8–22% depending on the use of different subscale combinations [22,28,46].

Rather than just finding out if a person gets symptoms associated with different chemicals, it would be important to examine how severe the symptoms are and how much the chemical intolerance affects the person’s life. In the previously mentioned study of Vuokko et al., 9.9% of the respondents also reported behavioral changes to avoid symptoms and 5.7% disabilities, e.g., disability to work, related to their sensitivities [31]. Respectively, a combination of the three QEESI© subscales (chemical intolerance, symptom severity, and life impact) could be a means of pointing out the most disabling cases of MCS in practice. Receiving high scores in all three subscales indicates that a person gets symptoms in association with several chemicals, has symptoms in different organ systems, and the symptoms considerably affect the person’s everyday life. In our study, the proportion of controls receiving high scores in all three subscales was 9%. Of the study patients, 26% received high scores in all three subscales indicating that a considerable proportion of their symptoms could be attributed to MCS.

Whether patients were diagnosed with asthma, asthma and/or chronic rhinosinusitis, laryngeal problem, or atopy or not did not influence MCS findings. This finding is contradictory to previous questionnaire studies reporting MCS being more common among subjects with respiratory tract inflammatory diseases and atopy [43,47,48]. MCS symptoms can, however, be interpreted as respiratory tract disease or allergic symptoms favoring, for example, diagnosis of asthma. It is worth noting that, in the present study, respiratory diseases were not diagnosed based on symptoms only, but asthma was diagnosed based on objective measures of lung function, CRS was diagnosed based on computed tomography and nasal endoscopy, and laryngeal disorders were based on indirect video laryngoscopy.

Women in the general population had more frequently scores referring to MCS compared to men. This finding is in agreement with previous studies [22,45,49], but there is no specific explanation for it. Women reporting more MCS may be linked to, e.g., women having a more sensitive olfactory function [50] or being more worried about possible health effects of environmental factors [51]. Among the study patients, women experienced more difficulties in everyday life because of the sensitivities than men, although there were no significant differences in chemical intolerance and symptom severity subscales. The reason for this is probably that the number of men in study patients was too small to produce statistical significance.

It is thought that MCS could develop after a single exposure event to a chemical (toxicant-induced loss of tolerance) or gradually [52]. Based on this cross-sectional study, it cannot be evaluated if MCS would have originated from MD exposure or if MCS is the primary reason for patients to have symptoms in an MD workplace. Either way, the possibility of MCS explaining at least a part of the patient’s symptoms associated with MD exposure at a workplace should be considered. Sufficient differential diagnostics considering the possible organic background of the patient’s symptoms is essential. In asthma treatment, the nature of respiratory symptoms requires thorough clarifying to avoid treating MCS symptoms with asthma medication. Since patients with IEI are a heterogenic group, careful multi-professional assessment of an individual patient’s situation and the background of the strain they usually have should be considered [53]. Palmquist suggested that psychotherapy aiming at reducing the emotional and behavioral reactions associated with exposure could be advantageous [37]. This seems reasonable as worrying and a negative affect may be connected to the development and permanence of MCS [13,54]. Mindfulness-based cognitive therapy [55], cognitive-behavioral therapy (CBT), or psychoeducation [56], however, have not so far proven to be efficient treatment choices in MCS, which may partly be explained by certain personality traits that some studies have linked with IEI [10]. There is also a possibility of MCS symptoms spontaneously recovering [37], and the knowledge of this could in part promote symptom relief. As regarding any diseases or symptoms, the nature of MCS symptoms should be well explained to the patient.

The strengths of this study are the systematic clinical examinations [44] of workplace-MD-exposed patients with the assessment of possible MCS with a questionnaire charting which chemicals or odors possibly cause symptoms, what kind of and how severe symptoms a person commonly experiences, and how the sensitivities affect different aspects of everyday life, and comparison of MCS results to the general working-age population. As seen in previous studies, MCS prevalence may vary depending on the target population within the same country, which is why the controls were selected to be working-age and from the same region as the patients lived in.

There are some limitations of the study. The response rate in the questionnaire for the controls in this study was quite low (38%), reflecting the willingness to take part in surveys in general nowadays. Even if the gender proportions in all and in different age groups of the study patients and controls were satisfactorily alike, those who are generally interested in the subject and perhaps more concerned about the effects of environmental factors on their health are probably more likely to take part in the survey, possibly causing the prevalence of MCS in the general population to be overestimated. This must be taken into account when interpreting the results, as the difference in the prevalence of MCS in the study patients and the general population may seem higher than it actually is. The set-up to compare MCS findings in a selected group of patients and the regional population can be questioned as there is limited knowledge on the background factors besides the age and gender of the controls. However, there is no information on, e.g., MCS in different occupations to favor inspection by occupation. Furthermore, considering the present conception of the mechanism of MCS, knowledge on the possible MD exposure of the controls is not essential. In addition, there is no knowledge of MCS/IEI prevalence among different patient groups in secondary health care.

To the best of our knowledge, QEESI© has been validated in the USA [21], Denmark [28], Japan [57], and Sweden [27]. It was chosen to be used in this study because of its properties enabling the evaluation of MCS difficulty and influence on everyday life and therefore seeming reliable to use in the assessment of MCS.

## 5. Conclusions

In conclusion, MCS is common among patients referred to secondary health care with respiratory tract and/or voice symptoms associated with workplace MD, and the symptoms considerably affect their everyday lives. MCS should thus be considered as a possible explanatory factor for MD-associated symptoms. MCS is common also in the general working-age population, although its prevalence may be overestimated in our study. In the future, follow-up research is needed to clarify the factors that explain the relief or worsening of MCS.

## Figures and Tables

**Figure 1 ijerph-18-12296-f001:**
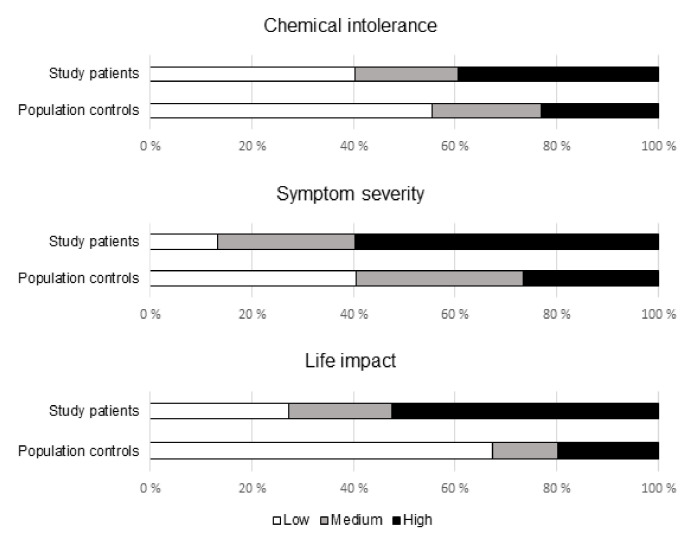
Proportions of subjects with low, medium, and high scores in chemical intolerance (*p* = 0.002), symptom severity (*p* < 0.001), and life impact (*p* < 0.001) among patients and controls (Χ^2^ testing with 3 × 2 crosstabulation).

**Figure 2 ijerph-18-12296-f002:**
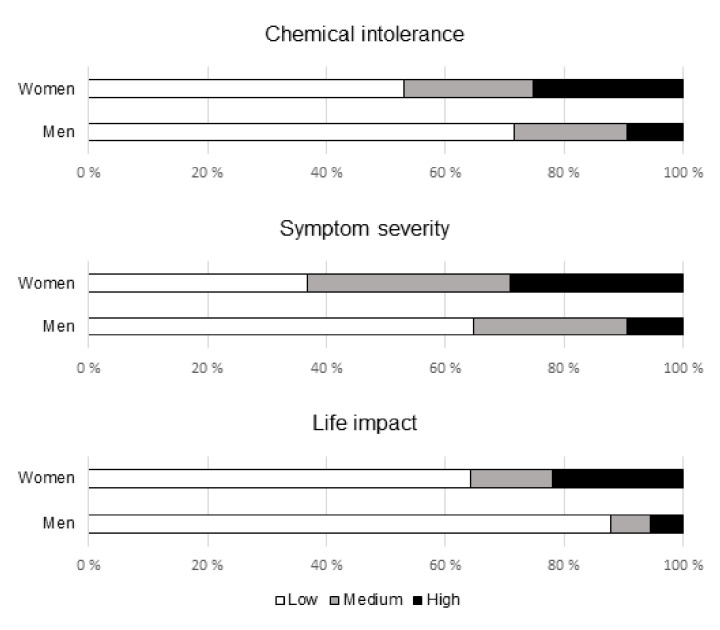
Proportions of controls with low, medium, and high scores in chemical intolerance (*p* = 0.004), symptom severity (*p* < 0.001), and life impact (*p* < 0.001) among women and men (Χ^2^ testing with 3 × 2 crosstabulation).

**Table 1 ijerph-18-12296-t001:** QEESI© questionnaire subscales used to assess possible MCS and the assessed items within each subscale.

Chemical Intolerance Subscale	Symptom Severity Subscale	Life Impact Subscale
Engine exhaust	Muscle or joint problems	Diet
Tobacco smoke	Eye or respiratory tract problems	Ability to go to work or school
Insecticides	Heart or chest problems	Furnishing home
Gasoline	Stomach or digestive system problems	Choice of clothing
Paint or paint thinner	Problems with ability to think	Ability to travel or drive a car
Cleaning products	Mood problems	Choice of personal care products
Perfumes or fragrances	Balance or coordination problems	Social activities
Fresh asphalt or tar	Headache or feeling of pressure in the head	Choice of hobbies and recreation
Nail polish, nail polish remover or hairspray	Skin problems	Relationship with spouse and family
New furnishings	Urinary tract or genital problems	Ability to clean home and perform other routine chores

**Table 2 ijerph-18-12296-t002:** Proportions of study patients with different illnesses or findings reporting high scores in chemical intolerance, symptom severity, and life impact subscales (CRS = chronic rhinosinusitis).

	Asthma (*n* = 32)			Asthma and/or CRS (*n* = 39)			Laryngeal Problem^1^ (*n* = 42)			Atopy (*n* = 37)		
Subscale	yes	no	*p*	yes	no	*p*	yes	no	*p*	yes	no	*p*
	**%**	**%**		**%**	**%**		**%**	**%**		**%**	**%**	
Chemical intolerance	44	37	0.661	42	36	0.675	48	33	0.207	30	45	0.143
Symptom severity	63	58	0.827	59	60	1.000	60	59	1.000	60	60	1.000
Life impact	50	54	0.830	46	57	0.410	56	52	0.837	51	53	1.000

^1^ Laryngeal dysfunction or organic change.

## Data Availability

The data presented in this study are available on request from the corresponding author. The data are not publicly available due to ethical restrictions.

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
