# Peer review of "Multiple Chemical Sensitivity in Patients Exposed to Moisture Damage at Work and in General Working-Age Population—The SAMDAW Study"

_ijerph, 2021, doi:10.3390/ijerph182312296_

Round 1

Reviewer 1 Report

The article “Multiple chemical sensitivity in patients exposed to moisture damage at work and in general working-age population – the SAMDAW study” by Nynäs et al. deals with an important clinical problem of multiple chemical sensitivity. It is a rather unknown to general practitioners syndrome and it seems that many patients do not get proper help. The paper deals with the prevalence of the symptoms, their severity and impact on patients’ all-day-life in general working-age Finnish population.

The Authors clearly present the data, however, the choice of the control group as described in line 107  could be better explained. Could the Authors comment on the difference, relationship, and impact on the population analyzed, between moisture damage and multiple chemical sensitivity? The link between these two phenomena is not clearly explained.

Author Response

Response: We thank the reviewer for taking the time to assess our manuscript and pointing out this important notion. We have added information on the control group. We have also added a paragraph on limitations of the study (rows 248-263), including discussion on the characteristics of the control group.

Reviewer 2 Report

This is an interesting study assessing multiple chemical sensitivity (MCS) in patients whose workplace exhibits moisture damage. The paper is well-written, the authors have followed a very good designed protocol based on the Quick Environmental Exposure and Sensitivity Inventory (QEESI) which is the state-of-art for the estimation of this condition.  A possible connection between moisture damage and MCS could be the exposure of patients to microbes that are abudant in such environment such as fungi (Hyvönen S, Poussa T, Lohi J, Tuuminen T. High prevalence of neurological sequelae and multiple chemical sensitivity among occupants of a Finnish police station damaged by dampness microbiota. Arch Environ Occup Health. 2021;76(3):145-151. doi: 10.1080/19338244.2020.1781034. Epub 2020 Jun 16). 

Author Response

Response: We appreciate the positive feedback from the reviewer. As presented in the introduction, MCS/IEI symptoms cannot be explained by any known toxicological, physical, or immunological mechanisms (rows 41-42). MCS can be temporally associated with MD exposure but as the key mechanism seems to be central sensitization the subject’s perception of noxious exposure (rather than true exposure) is essential in the development of MCS. Again, we thank the reviewer for taking the time to assess our manuscript.

Reviewer 3 Report

Well written manuscript. 

To the point and original analysis regarding the damage (MD) associated respiratory tract or voice symptoms and multiple chemical sensitivity (MCS).

The material and methods are clear and the results reporting are coherent and sound scientifically. The discussion is adequate and contains an indepth analysis of the findings. 

Suggestion: I would add a paragraph on the limitation of the study in the discussion section discussing: the downside of using surveys, the time lag between the exposure and the outcomes, and the bias that could be created by the referral to secondary health care.

Author Response

Response: We appreciate the positive feedback from the reviewer. We thank the reviewer for the suggestion to add a paragraph on limitations of the study, which we have now done (rows 248-263). However, we would like to point out that this study did not seek causality between MD exposure and MCS. Our aim was to investigate the presence of MCS among patients referred to secondary health care and compare it to the prevalence of MCS in the regional population. That is why the time lag between the MD exposure and the outcomes is not relevant here. Again, we thank the reviewer for taking the time to assess our manuscript.